# Optimizing First-Line Chemotherapy in Metastatic Pancreatic Cancer: Efficacy of FOLFIRINOX versus Nab-Paclitaxel Plus Gemcitabine

**DOI:** 10.3390/cancers15020416

**Published:** 2023-01-08

**Authors:** Francesco Di Costanzo, Federica Di Costanzo, Lorenzo Antonuzzo, Ernesto Mazza, Elisa Giommoni

**Affiliations:** 1Medical Oncology Unit, Hospital Villa Donatello, 50019 Florence, Italy; 2Medical Oncology Unit, Hospital Santa Maria della Stella, 05018 Orvieto, Italy; 3Medical Oncology Unit, Careggi University Hospital, 50134 Florence, Italy; 4Interventional Radiology Unit, Hospital Villa Donatello, 50019 Florence, Italy

**Keywords:** metastatic pancreatic cancer, nab-paclitaxel, FOLFIRINOX

## Abstract

**Simple Summary:**

Metastatic PC often represents a complex decision-making moment for oncologists, particularly due to the general conditions of the patients, age, and the potential toxicity of chemotherapy. Choosing the best first-line treatment is the first step towards improving survival and quality of life. In this review, the authors discuss the most accredited strategies today and the predictable toxicities, as well as clinical factors (age, PS, etc.).

**Abstract:**

Pancreatic cancer (PC) is one of the most lethal tumors in Europe with an overall 5-year survival rate of 5%. Since 1992, gemcitabine (Gem) has been the treatment of choice for metastatic disease with significant improvement in median overall survival (OS) compared to fluorouracil. A good performance status (PS) at diagnosis appears to be a strong predictive factor for better survival. Overall, 50% of PC are metastatic or locally advanced at diagnosis, and more than 70% of the resected patients will experience a recurrence, with a median OS ranging from 4 to 10 months (mos). FOLFIRINOX (5-fluorouracil, leucovorin, irinotecan, and oxaliplatin) and Nab-paclitaxel (Nab-p) plus Gem have recently increased survival of patients with metastatic PC, over Gem. Treatment with FOLFIRINOX is generally considered more effective with respect to the doublet, with toxicity concerns, FOLFIRINOX achieves an overall response rate (ORR) of 31.6%, while for Nab-p plus Gem ORR is 23%; however, FOLFIRINOX was associated with higher rates of grade 3 and higher adverse events. Although the international guidelines indicate that both regimens can be used as first-line therapy for patients with metastatic PC, FOLFIRINOX is the most widely used; Nab-p plus Gem is more frequently used in patients with lower PS. In this review, we critically analyze these two regimens to give a pragmatic guide to treatment options.

## 1. Introduction

In Europe, pancreatic cancer (PC) is the fourth cause of cancer death in both sexes with approximately 140,000 new cases per year and almost the same number of deaths [1]. PC death rates since 2000 continue to slowly increase in men (0.3% annually) and remain stable in women [2].

Fifty percent of new diagnoses are in an advanced stage (locally advanced or metastatic) and more than 70% of the resected patients will experience a recurrence, with median overall survival (OS) ranging from 4 to 10 months (mos) [3,4]. The estimated median survival from diagnosis without treatment for metastatic (m) PC, based on large case series and national registries, is 3 mos in a US study of 32,452 patients and 10 weeks in a Dutch study of 3099 patients, mainly due to the difficulties in the diagnosis for the deteriorated PS at diagnosis and the lack of effective treatments [2,5]. Carrato et al. reported in a systematic review that the median OS in patients with metastatic PC in Europe was less than 5 mos and that less than 10% of patients survived beyond 5 years [6].

Approximately 70% of PC arise in the head of the pancreas, often causing biliary obstruction, leading to jaundice, appetite and weight loss, fatigue, and exocrine pancreatic insufficiency. In patients with body and tail pancreatic cancers, nonspecific symptoms are frequent, including abdominal and back pain, appetite loss, weight loss, and fatigue.

Standard therapy for advanced or mPC is chemotherapy (CT), which improves survival compared with best supportive care according to a meta-analysis of seven randomized trials. For more than a decade, gemcitabine (Gem) was the approved single agent treatment, while all other phase III trials evaluating gem-based combination over mono CT failed to significantly improve OS, before the MPACT trial [7,8]. 

Storniolo and co-workers reported in an investigational new drug treatment program that patients with a Karnofsky performance status (KPS) ≥ 70% had a median OS of 5.5 mos versus only 2.4 observed in patients with KPS < 70%, and median time to disease progression (TTP) was better in patients with good PS [9]. This trial supported the value of PS as a surrogate endpoint of survival and TTP. Several trials confirmed baseline PS as an important prognostic factor to consider for developing a therapeutic strategy. Patients with a KPS below 70% or PS ECOG more than 1 have a marginal benefit with mono-CT alone. This aspect, in light of the recent introduction of multi-drug combinations, plays a very important role in the selection of patients and interpretation of the results.

Relevant improvement in the first-line treatment of metastatic PC was provided by FOLFIRINOX (5-fluorouracil [5-FU], leucovorin [LV], irinotecan [CPT11], and oxaliplatin [Oxa]) and Gem plus nab-paclitaxel (Nab-p) [10,11]. In Table 1, we report the main treatments most frequently used in metastatic PC and the achieved results. 

An important aspect in the treatment of mPC, often underestimated by clinicians, concerns the quality of life (QoL) and the use of simultaneous supportive care. QoL data are very few, but in some cases suggest a benefit of CT over best supportive care (BSC), particularly in case of achievement of objective response with symptoms control [13]. Early supportive care would be considered at the beginning of treatment for advanced PC to improve treatment compliance and QoL, as well as control symptoms and avoid toxicities, allowing for better results, especially in terms of survival.

In this review, we want to critically analyze the role of multi-drug treatments in the first line for mPC in order to give a pragmatic guide to treatment options.

## 2. FOLFIRINOX

At present, one of the current standard first-line regimens for patients with advanced disease includes the combination FOLFIRINOX.

The rationale for using the FOLFIRINOX combination in PC is mainly based on the demonstrated therapeutic activity of irinotecan in mPC [14,15] and the evidence that the combination of irinotecan with LV and fluorouracil has significant synergistic activity; furthermore, Oxa has significant clinical activity in PC only when combined with fluorouracil and also has a synergistic effect in vitro when combined with irinotecan [16,17,18,19,20,21].

The French intergroup trial (PRODIGE 4/ACCORD 11) investigated the FOLFIRINOX combination in a randomized study versus Gem alone. This combined treatment achieved a median OS of 11.1 vs. 6.8 mos (HR 0.57; *p* < 0.001) for Gem and a median progression-free survival (PFS) of 6.4 compared to 3.3 mos (HR 0.47; *p* < 0.001) for Gem alone [9]. Subsequent real-life experiences and cohort studies confirmed these results [22,23,24,25,26].

Only 38% of patients in the FOLFIRINOX trial had pancreatic head cancer compared with other studies reporting inclusion rates of 52 to 70% [27,28] (Table 2). This is due to the enrollment criteria, which excluded elevated bilirubin levels due to the increased risk of irinotecan-induced toxicity [29].

Synchronous metastases, a low baseline albumin level (<3.5 g per deciliter), liver metastases, and age greater than 65 years have been identified as independent negative prognostic factors for OS.

However, FOLFIRINOX showed significant toxicities with higher rates of grade ≥3 neutropenia with respect to Gem alone [11]. The toxicity of the three-drug FOLFIRINOX regimen was superior to that of Gem with a higher rate of grade 3/4 neutropenia (45.7% vs. 21%), febrile neutropenia (5.4% vs. 1.2%), and diarrhea (12.7% vs. 1.8%). In addition, approximately 42% of patients treated with FOLFIRINOX required supportive therapy with G-CSF compared with only 5.3% in the Gem group. 

Based on the high toxicity, in real-life clinical use, it is reported that a modified (m) FOLFIRINOX schedule, omitting the fluorouracil bolus and reducing the irinotecan dose from 180 to 150 mg/m^2^, was widely used.

Kang et al. and Ghorani et al. reported that mFOLFIRINOX has comparable efficacy to original FOLFIRINOX in patients with mPC, with less toxicity [30,31]. Gunturu K. et al., in a meta-analysis, reported the same conclusions in terms of survival benefits and reduction of toxicity compared with the original schedule [32].

Only patients younger than 75 years old were included in the PRODIGE-4 study, therefore, there were no data for older subjects regarding efficacy and toxicity in elderly patients.

Garcia et al. and Mizrahi et al. recently published a review on the use of new regimens of chemotherapy in elderly patients with advanced PC [33,34]. In their conclusions, they reported that fit elderly patients may have a similar therapeutic benefit for patients younger than 75 years old, however, they suggested a reduction in the original schedule dose and support with G-CSF as primary prophylaxis. A retrospective study in metastatic PC evaluated 88 patients with a median age at diagnosis of 56 years old (range 32–78 years old) treated with FOLFIRINOX in accordance with the PRODIGE regimen and showed no clear differences in outcomes and toxicity [35].

The Italian Cooperative Oncology Group for Clinical Research (GOIRC) in phase I/II trial evaluated the replacement of either Oxa or CPT11 with Nab-p in the original FOLFIRINOX schedule, obtaining two new doublets: Nab-FOLFIRI(5-FU+LV+CPT-11) (arm A) and Nab-FOLFOX (5-FU+LV+OXA) (arm B) [36]. The primary objective of phase 1 was the definition of the maximum tolerated dose (MTD), while for phase II the safety and activity of Nab-FOLFIRI and Nab-FOLFOX in m PC. Sixty-three patients received Nab-FOLFIRI or Nab-FOLFOX in phase I, and MTDs were defined at 120 mg/m^2^ for Nab-p with FOLFIRI and 160 mg/m^2^ with FOLFOX. In phase II, 42 patients for each arm were randomized. The results showed that the overall response rate (ORR) was 31% for both schedules; the clinical benefit rate (CBR) was 69% and 71%, 1-year survival was 41% and 50%; PFS was 6 and 5.6 mos, and median OS was 10.2 and 10.4 mos for Nab-FOLFIRI and Nab-FOLFOX, respectively. Neutropenia was the most common grade ≥ 3 adverse event in both regimens, significantly lower than that reported for the FOLFIRINOX triplet. Triplets containing Nab-p might be hopeful first-line CT options for metastatic PC patients if further investigated. 

In a subsequent French randomized phase II study (PANOPTIMOX), the authors evaluated the duration of treatment as an endpoint, comparing FOLFIRINOX for 12 cycles versus FOLFIRINOX for 8 cycles, followed by maintenance treatment LV and 5FU versus a sequential treatment alternating Gem and FOLFIRI (FIRGEM) every 2 mos. The resulted yielded a median OS of 10.1, 11.2, and 7.3 mos, respectively, demonstrating that the maintenance strategy can be superimposed on continuous FOLFIRINOX [37]. 

Reure et al. reported that after 4–8 cycles of FOLFIRINOX, it was possible to switch to maintenance treatment with capecitabine alone, achieving a median OS of 17 mos and a median PFS of 5 mos [38]. Frank et al. obtained interesting results in terms of PFS with maintenance on the FOLFIRI regimen after 2–6 mos of induction treatment with FOLFIRINOX [39]. In this cohort of 22 patients, the median overall PFS was 11 mos. A further study by Hann et al. showed a PFS of 10.6 mos (95% CI, 6.7–14.4) and OS of 18.3 mos (95% CI, 14.8–21.8) in a cohort of 13 patients whose maintenance was established with 5FU alone after the FOLFIRINOX regimen [38,40,41].

All AAs evaluating maintenance after at least four cycles of FOLFIRINOX induction chemotherapy observed that both patients with stable disease and those with objective responses had similar survival outcomes, suggesting that de-escalation of FOLFIRINOX with 5FU, capecitabine, or FOLFIRI was an optimal strategy after disease control was achieved. These options make it possible to reduce the cumulative toxicity of FOLFIRINOX and open the possibility, as in advanced colorectal cancer, to hypothesize a reinduction strategy in mPC. Regarding maintenance treatment, we have to consider that 4–8% of PC patients have germline mutations of BRCA1 or BRCA2 [39]. Both these mutations are implicated in the encoding of proteins of the homologous repair pathway and those of the rupture repair of the double strand of DNA [42,43]. This group of patients with BRCA1/BRCA2 mutations has a greater response to first-line chemotherapy with platinum-based regimens. Poly (ADP-ribose) polymerase (PARP) inhibition is hypothesized to synergistically work with BRCA1/BRCA2 mutations to inhibit single-strand break repair. PARP inhibitors have been used in the therapy of metastatic PC with BRCA1/BRCA2 mutations as response maintenance therapy after chemotherapy induction [44]. 

## 3. Nab-Paclitaxel and Gemcitabine

Nab-paclitaxel ([Abraxane], Celgene Corporation, Summit, NJ) is an albumin-bound paclitaxel that has been demonstrated to improve survival outcomes in metastatic breast cancer and non-small cell lung cancer in several clinical trials [45,46].

This formulation, without the use of solvents that keep paclitaxel soluble once injected, has a better safety profile with respect to paclitaxel in terms of allergic reactions.

In vivo studies in mice injected with human pancreatic tumor cells showed that the combination of Gem and Nab-paclitaxel was more effective than the single drugs alone.

Subsequently, phase I-II studies showed interesting results in 2011 also in patients with mPC, in particular when it was combined with Gem: median OS of 12.2 mos, RR of 48%, and a median PFS of 7.9 mos [47]. Gemcitabine is one of the most active single chemotherapy drugs in advanced PC with a median OS of 5.7 mos and a 1-year survival of 20% [48]. Prior to the acquisition of the latest results in the treatment of mPC, Gem as a single agent has long been the standard of care. Today, Gem monotherapy is indicated in frail subjects with poor PS and comorbidities. In several randomized trials, it was inferior to two more recent chemotherapy combinations: FOLFIRINOX and Nab-p plus Gem.

The cellular uptake of Gem is related to the activity of the human equilibrated nucleoside transporter (hENT1) protein [49]. The prospective randomized study of adjuvant therapy RTOG9704 evaluated hENT1 immunohistochemistry [48,50]. In univariate and multivariate analyses, elevated hENT1 expression was associated with improved OS and DFS in the Gem group (HR, 0.40; *p* = 0.004; and HR, 0.39; *p* = 0.003) but not in the group given 5-FU.

The international phase III MPACT study randomized 861 patients with mPC to an innovative schedule that included Nab-p plus Gem (Nab-p: 125 mg/m^2^ plus Gem: 1000 mg/m^2^ on days 1, 8, and 15 of each 28-day cycle) versus Gem alone [10]. The combination achieved statistically significant results in all major endpoints considered, compared with Gem alone: OS of 8.5 vs. 6.7 months, (HR = 0.72, 95% CI = 0.62 to 0.83, *p* < 0.001), median PFS of 5.5 vs. 3.7 months (HR = 0.69, 95% CI = 0.58 to 0.82, *p* < 0.001), and ORR of 23% vs. 7% (response rate ratio = 3.19, 95% CI = 2.18 to 4.66, *p* < 0.001), respectively. In the combination arm, the most frequent grade ≥3 adverse events were neutropenia, leukopenia, fatigue, and peripheral neuropathy.

An update of MPACT data, after a median follow-up of 13.9 mos with 774 deaths (90% of patients), including 380 patients in the combination group (88%) and 394 in the Gem alone group (92%), confirmed that the median OS for patients who received Nab-p plus Gem was 8.7 mos (95% CI = 7.89 to 9.69), which was statistically significantly longer than Gem alone (6.6 mos (95% CI = 6.01 to 7.20, HR = 0.72, 95% CI = 0.62 to 0.83, *p* < 0.001)) [51]. Most patients had liver metastases (84%) and over 90% had more than one metastatic site. The absence of liver metastases was associated with better OS in the Nab-p plus Gem arm (11.1 vs 8.3 mos, HR = 0.58, *p* = 0.001) and in the Gem arm alone (10.2 vs 5.9 mos, HR = 0.58, *p*< 0.001). In this update, it was also possible to intercept 4% of patients who survived longer than 36 mos and 3% of patients who survived at least 42 mos in the combination arm. No patient survived for more than 36 mos in the Gem treatment group. The update confirmed the efficacy of the combination in most of the parameters analyzed and in the patient subgroups. Multivariate analysis showed that visceral metastases and KPS were statistically significant independent predictors of survival.

In the MPACT study, a Cox regression analysis using study stratification factors as covariates to identify predictors of OS revealed that both the KPS score and the presence or absence of liver metastases were independent predictors of survival.

CA19-9 has been shown to be a univariate predictor of OS in PC, with high values associated with poor survival [52]. Patients treated with Nab-p plus Gem achieved similar OS regardless of whether baseline CA19-9 levels were below, equal to, or greater than the median, whereas in the Gem arm, OS was statistically significantly better for subjects with baseline CA19-9 levels below the median compared with those with levels greater than or equal to the median (*p* = 0.001).

These data suggest that for the Nab-p plus Gem regimen, Ca19.9 has no predictive value, whereas it is a predictive factor for Gem alone and may be related to the fact that the combination acts on metabolic pathways not amenable to blockade by Gem alone [53].

Unlike FOLFIRINOX, the Nab-p plus Gem combination was also indicated in patients with advanced PC with a KPS of 70 to 80 in whom no grade 3 or higher adverse events were observed.

In an Italian retrospective real-life experience, 105 patients treated with Nab-p plus Gem were included [54]. In the subgroup aged ≥ 70 years old (37 patients) the median of cycles received was inferior (5 cycles, range 1–10) for younger patients (median 6 cycles, range 1–12), DCR was 48% (9 PR + 9 SD), and no differences in term of PFS (6.5 mos, 95% C.I. 5.36–7.64, *p* = 0.49) and OS (10 mos, 95% C.I. 8.53–11.47 *p* = 0.67) were detected for < 70 years old patients. In the older population, authors reported more G3-4 non-hematological (27% vs 15% *p* = 0.03) and fewer hematological (12% vs 29% *p* = 0.004) events for younger patients. 

To date, Nab-p plus Gem is administered until progression, and a depowering of treatment, such as maintenance, is not standardized. Blomstrand et al., in a real-life experience with locally advanced or mPC, registered dose reductions in 80% of patients due to cumulative toxicities [55]. 

The systematic review followed the recommendations of the Preferred Reporting Items for Systematic Reviews and Meta-Analyses (PRISMA). The protocol has not been registered.

## 4. Discussion

To date, no randomized clinical trials have been conducted to compare the efficacy of FOLFIRINOX versus Nab-p plus Gem. Furthermore, clinical trials often select patients who do not represent the real world and thus limit decision making [56]. For example, regarding age and PS, the FOLFIRINOX study only included patients under the age of 75 and 99% of patients with ECOG PS 0-1, while in the MPACT trial, 40% of patients had a KPS between 80 and 60%, and 42% of patients were over 65 years.

Chiorean I et al. in a meta-analysis evaluated 34 studies involving a total of 6915 patients with mPC treated with Nab-p plus Gem or FOLFIRINOX [57], data showed an OS of 14.4 and 15.9 mos with Nab-p plus GEM and FOLFIRINOX and an mPFS of 8.5 and 11.7 mos, respectively. Toxicity analysis in safety data was reported in 14 studies (2205 patients), finding higher rates with FOLFIRINOX versus Nab-p plus Gem for grade 3/4 neutropenia and febrile neutropenia, while rates of peripheral neuropathy of grade 3/4 were higher with Nab-p plus GEM in four of the seven studies. The authors of the meta-analysis reported a numerically greater percentage of patients who discontinued FOLFIRINOX treatment due to adverse events in two studies, while they reported that grade 3/4 rates of peripheral neuropathy were higher with Nab-p plus GEM compared with those of FOLFIRINOX. 

In the MPACT study, Nab-p plus GEM-associated grade ≥3 neuropathy was reported to improve to grade 1 in a median of 29 days. Studies reporting supportive treatments reveal lower use of G-CSF with Nab-p plus GEM.

Gresham et al., in a systematic review and meta-analysis including 23 randomized clinical trials with chemotherapy (Nab-p plus Gem or FOLFIRINOX) in patients with advanced PC (9989 patients), found no clear difference in efficacy between the two regimens in terms of OS or PFS. FOLFIRINOX was associated with a higher incidence of grade 3/4 neutropenia with respect to Nab-p and Gem [58].

Assenat E. et al. evaluated the alternation of FOLFIRINOX and Gem/Nab-p, obtaining very promising results in a phase II exploratory study with an ORR rate of 63%, mPFS of 10.5 mos (95% CI 6.0–12.5 mos), and a mOS of 15.1 mos (95% CI: 10.6–20.1 mos) in 58 patients [59]. 

Based on the results of the ACCORD and MPACT studies, FOLFIRINOX seems to perform better than Nab-p plus Gem for median OS (11.1 vs. 8.5 mos), median PFS (6.4 vs. 5.5 mos), and ORR (31.6 vs. 23%), respectively, and is therefore preferable to the doublet; however, the triplet showed higher toxicity rates, with increased high-grade neutropenia (45.7 vs. 38%), fatigue (23.6 vs 17%), and diarrhea (12.7 vs. 6%). 

If we compare resource utilization and cost of care for patients with PC treated on the first line with FOLFIRINOX or Nab-p plus Gem, we must note that the triplet has a lower overall cost but requires greater economic resources for the supportive therapy for toxicity [60]. Overall, the OS of most studies showed an advantage in favor of the combination with FOLFIRINOX over Gem plus Nab-p while other studies demonstrated comparable efficacy [61]. 

In 2022, Klein-Brill et al. performed a retrospective, nonrandomized study comparing FOLFIRINOX and Gem plus Nab-p in mPC using administrative data from the AIM Specialty Health-Anthem Cancer Care Quality Program and administrative claims of commercially insured patients to describe OS and post-treatment costs and hospitalization [62]. A total of 1102 patients were included: those treated with FOLFIRINOX were younger (median age, 59.1 vs. 61.2 years; *p* < 0.001), with better PS (226 ([39.9%])) vs. (176 [32.8%]) than the patients treated with Gem plus nab-p. The median OS was 9.27 and 6.87 mos for FOLFIRINOX and Gem plus Nab-p, respectively (*p* < 0.001). The survival benefit was seen across all subgroups, including different ECOG PS, ages, and the number of metastatic sites. The FOLFIRINOX arm also had 17.3% fewer post-treatment hospitalizations (*p* = 0. 03) and a 20% lower post-treatment cost reduction (*p* < 0.001).

Reported toxicity assessments are often inconsistent, no apparent difference in adverse effects was reported in some studies. In some experiences with Gem plus Nab-p, more adverse events were present with consequent treatment interruption, while others show increased toxic effects in patients treated with FOLFIRINOX [63].

Beggs et al. reported in an evaluation of human PC cell lines that there is a differential chemo-sensitivity to FOLFIRINOX and Gem plus paclitaxel, demonstrating the heterogeneity of response to these drug regimens compared with the cell lines studied [64]. Six of nineteen cell lines showed optimal sensitivity to FOLFIRINOX, while three of nineteen showed optimal sensitivity to Gem-Pac. Two cell lines had similar sensitivities to the two regimens, while the other cell lines did not respond to either chemotherapy regimen. Cell lines that were optimally responsive to one drug regimen were not to the other, indicating that primary choice plays a role. This study preliminary suggests that since the overall percentage of FOLFIRINOX-sensitive cell lines compared with Gem-Pac-sensitive cell lines is higher, the former regimen has more first-line indication, although only a randomized study could corroborate these data.

Most mPC carry a mutation in the key driver K-ras, and over half have mutations and/or copy number losses of TP53, SMAD4, and CDKN2A [43,65,66]. Since about 30% of PC patients have important structural genomic aberrations, mutational signatures, and RNA expression profiles (basal-like), we can hypothesize that this subgroup of mPC will benefit from personalized therapeutic approaches in the near future, while for patients with the “classical” mPC RNA subtype, chemotherapy is currently the standard treatment. Most of these mutations are more frequent in metastatic disease [67,68].

Aung et al. have recently demonstrated that in patients with the “classical” subtype of mPC treated with m-FOLFIRINOX, it is possible to obtain a better ORR and PFS than first-line chemotherapy compared with “basal-like patients” [69]. GATA6 expression, as assessed by RNA in situ hybridization technique, was found to be an excellent surrogate biomarker for differentiating “classical” and “basal-like” mPC subtypes [70].

## 5. Conclusions

This brief review provides an analysis of the clinical results obtained over the past two decades to improve treatment outcomes of mPC. Although numerous combinations of chemotherapy and biologics have been evaluated, Gem has been the most effective treatment since 1992; subsequently, FOLFIRINOX or Nab-p plus Gem have been established as alternative first-line combinations for patients with mPC, although the impact on OS was marginally successful. 

Great care must always be taken when comparing the data of two different trials, firstly because this is without any statistical and methodological rule, and secondly because the inclusion criteria and other parameters influenced the composition of the populations of different studies. For example, the FOLFIRINOX study was conducted in a cooperative setting in a single country (France), while the MPACT study is an international study sponsored by a company.

In our experience, the FOLFIRINOX regimen is an optimal treatment for patients with mPC with good PS. The mFOLFIRINOX regimens achieve comparable results with a reduction in systemic toxicity. In real-life practice, it is routinely used, albeit in absence of strong evidence, as a maintenance strategy after 8–12 cycles of mFOLFIRINOX, which can minimize cumulative toxicities. In patients with germline mutation of BRCA1 and BRCA2, platinum-based CT is the regimen of choice; therefore, in countries where regulatory rules make it possible, FOLFIRINOX is preferable over Gem plus Nab-p because there is the possibility of using maintenance with olaparib [70]. 

In many European countries, the choice of using the doublet in the first line is guided by regulatory rules because Nab-p can only be used in a first-line setting. 

Nab-p plus Gem has demonstrated comparable outcome results, however, in the past, the cost of Nab-p limited its use in some countries. Our experience in a phase I-II study evaluated the modification of the original FOLFIRINOX schedule with the replacement of Oxa and Iri with Nab-p (Nab-FOLFOX and Nab-FOLFIRI), obtaining interesting results, but to date, phase III trials are not planned.

The therapeutic choice should also take into account the PS and age. In subjects with low PS and over 75 years of age, the doublet can be better tolerated, but in some cases, it is possible to consider treatment with Gem alone.

In conclusion, in the overall decisions about the upfront therapeutic strategy for metastatic PC, a careful and multidimensional assessment of the patient at baseline is mandatory with a multidisciplinary approach and early start of supportive care, including nutritional support and a geriatric evaluation at baseline for elderly patients 

Furthermore, clinicians should always take care of patients’ needs, expectations, and provide a clear explanation of the goals of treatment, prognosis, and probable toxicities.

Prospective genomic profiling of mPC will improve patient selection and therapeutic outcomes.

## Figures and Tables

**Table 1 cancers-15-00416-t001:** First-line regimes in metastatic PC.

Regimen	Phase	*n* Patients	mPFS (mos)	mOS (mos)	RR	Ref.
FOLFIRINOX vs	III	342	6.4	11.1	70	[11]
gemcitabine			3.3	6.8	51	
Nab-p + gemcitabine vs	III	861	5.5	8.7	48	[10]
gemcitabine			3.7	6.6	33	
Gemcitabine vs	III	126	9 wks *	5.6	5.4	[12]
5-FU			4 wks *	4.4	0	

mPFS: median progression free survival; mOS: median overall survival; RR: response rate; * time to progression.

**Table 2 cancers-15-00416-t002:** Principal characteristics of patients included in FOLFIRINOX and MPACT trials.

Characteristic	FOLFIRINOX	Gem	Nab-p+ Gem	Gem
Median age (range)	61 (25–76)	61 (34–75)	62 (27–86)	63 (32–88)
Sex: M/F (%)	62/38	61.4/38.6	57/43	60/40
Best PS (%) (ECOG PS 0-1 or KPS 100-90)	99.3	100	58	62
Pancreatic primary tumor location				
Head	39.2	36.8	44	42
Body	31.0	33.9	31	32
Tail	26.3	26.3	24	26
Unknown	0	0	1	1
Number of sites involved (range)	2 (1–6)	2 (1–6)	NA	NA
Sites of metastases (%)				
Liver	87.6	87.7	85	84
Lung	19.4	28.7	35	43
Peritoneum	19.4	18.7	4	2

## Data Availability

The data presented in this study are available in this article.

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
