# Peer review of "Optimizing First-Line Chemotherapy in Metastatic Pancreatic Cancer: Efficacy of FOLFIRINOX versus Nab-Paclitaxel Plus Gemcitabine"

_cancers, 2023, doi:10.3390/cancers15020416_

Round 1
Reviewer 1 Report
Cancers (ISSN 2072-6694)
COMMENTS FOR THE AUTHOR:
The manuscript “Optimizing first-line chemotherapy in metastatic pancreatic cancer: FOLFIRINOX or Nab-paclitaxel plus gemcitabine?” by Francesco Di Costanzo et al. is a concise and fascinating review. This review has reasonably provided an analysis of the clinical results obtained over the past two decades to improve treatment outcomes of metastatic PC specifically, using two regimens FOLFIRINOX (5-fluorouracil; irinotecan and oxaliplatin) and Nab-paclitaxel (Nab-p) plus Gem.
Overall, it provides an update on the current scientific literature related to this area. However, the manuscript needs some refinement and correction of a few errors, as mentioned below:
Minor comments:
The authors should discuss the research gap in this field, particularly future perceptions of using these combinatorial drugs. It would be a significant segment for readers and researchers regarding future research perspectives.
The authors need to incorporate the graphical/ bar/ survival curve using Tables 1 and 2 or the bioinformatics tool; it would be more appealing for understanding.
Reference citations are missing; for example (lines no- 40- 43; 75-76; 102-104; 109-111 etc.). Please have a look over it and revise the manuscript carefully.
Generally speaking, the manuscript is written in good English, while there are several typos, grammatical errors, and fragmentation of sentences. Please carefully revise and correct the mistakes.
Reviewer 2 Report
Summary:
The authors review the current literature regarding the use of gemcitabine plus nab-paclitaxel vs FOLFIRINOX in patients with metastatic pancreatic cancer. They highlight key trials that demonstrate the efficacy of both regimens over monotherapy. Based on their review, they recommend FOLFIRINOX in patients with good PS ECOG and gemcitabine in elderly patients or those with low PS. The authors understand that RCTs will be required to further delineate the utility of each regimen.
General/Specific Comments:
1. "For more than a decade, gemcitabine (Gem) was the approved single agent treatment, while all other phase III trials evaluating gem-based combination over 47 mono CT failed to improve OS significantly". The wording of this seems confusing as, a few lines later, a phase III trial is cited demonstrating the OS benefit with gem +nab-p over gem mono.
2. While the summary of the studies referenced in this article is acceptable, it is still simply a summary. There is no additional review, meta-analysis, or new insight. Furthermore, discussions regarding chemotherapy choice in PDAC should also delve into the molecular subtypes of PDAC (e.g. classical, squamous) and how that influences response to these regimens.
Author Response
Please see the attachement

Reviewer 3 Report
This manuscript by Di Costanzo et al. presents a comprehensive review of the clinical use of FOLFIRINOX (5-fluorouracil; irinotecan and oxaliplatin) and Nab-paclitaxel (Nab-p) plus Gemcitabine (Gem) in first-line treatment of metastatic Pancreatic cancer (mPC). The review is objective, and the conclusions are based on real-life clinical data. And provides clinicians a pragmatic guide for the choice of therapy based on the PC patient's conditions. The review is timely and should capture the interest of the wide pancreatic audience. There are a few grammatical errors that should be addressed.
A reference should be provided for the statement on lines 172-173.
Author Response
Please seee the attachment

Round 2
Reviewer 2 Report
1. To clarify again, the query was regarding the statement concerning gem vs gem combo therapy. There was slight confusion in that gem mono was initially mentioned as being superior to gem combo, while trials are cited later in the text indicating otherwise. But it seems that the point was that in earlier decades gem mono was superior.
2. In making this a comprehensive analysis, I would still recommend including statements regarding PDAC subtypes. The variants mentioned in the response are other subtypes of pancreatic cancer. There are molecular subtypes of PDAC that can influence the type of chemotherapy that the PDAC is sensitive to and there are ongoing transcriptome trials evaluating this. See PMID 34805152 as an example.
Author Response
"PLease see the attchement"
